# Burden of Atopic Dermatitis in Adults in Greece: Results from a Nationwide Survey

**DOI:** 10.3390/jcm11164777

**Published:** 2022-08-16

**Authors:** Stamatis Gregoriou, Garyfallia Stefanou, Stathis Kontodimas, Konstantinos Sfaelos, Maria Zavali, Efstratios Vakirlis, Georgia Kourlaba

**Affiliations:** 1Faculty of Medicine, Andreas Sygros Hospital, National and Kapodistrian University of Athens, 16121 Athens, Greece; 2ECONCARE, 11528 Athens, Greece; 3LEO Pharma Hellas, 14564 Athens, Greece; 4Department of Wine, Vine & Beverage Sciences, University of West Attica, 12243 Egaleo, Greece; 5First Department of Dermatology and Venereology, School of Medicine, Aristotle University, 54622 Thessaloníki, Greece; 6Department of Nursing, University of Peloponnese, 22100 Tripoli, Greece

**Keywords:** survey, atopic dermatitis severity, quality of life, work productivity, absenteeism, sleep loss, health resources

## Abstract

The objective was to describe the AD burden in terms of quality of life (QoL), sleep, social life, work productivity, and resource utilization in Greece and assess the impact of disease severity. A nationwide cross-sectional survey was conducted. The questionnaire consisted of socioeconomic factors, medical history, AD screening, AD severity, QoL, sleep difficulties, social activities, and work productivity questions. AD was defined using the UK Working Party criteria (UKWP cohort) and a patient-reported AD diagnosis from a physician (Expert Diagnosis cohort). Self-reported moderate/severe AD was estimated using the Patient-Oriented Eczema Measure (POEM). In the UKWP cohort, the AD effect on QoL was moderate/extremely large in 84.3% of moderate/severe AD (vs. 55.7% in mild; *p* = 0.016), while in the Expert Diagnosis cohort, it was 72.2% (vs. 22.8%; *p* < 0.001). Disease severity was associated with a higher impact on sleep and social activities. Overall work impairment was high in both mild (32.7%) and moderate/severe (48.5%) AD of the UKWP cohort, while among the Expert Diagnosis cohort, it was significantly higher among those with moderate/severe (31.2%) versus mild AD (11.9%; *p* < 0.001). The AD burden in Greece is significant, especially for those in moderate/severe AD stages. Acknowledging this burden is the first step toward applying healthcare decisions that will benefit patients and the community.

## 1. Introduction

Atopic dermatitis (AD) is a common, chronic, and relapsing inflammatory disease that is characterized by intense itch and dry skin [1]. The annual prevalence of doctor-diagnosed AD in adults ranges from 1.2% in Asia to 17.1% in Europe based on a recently conducted systematic literature review [2]. In a large cross-sectional multinational study recently published by Barbarot et al. (2018), the annual prevalence of adult AD based on self-reporting of symptoms was 4.3% in Japan, 8.1% in Canada, 9.4% in Europe, and 11.9% in the US [3]. In another cross-sectional study conducted in five European countries (Germany, Italy, the Netherlands, Portugal, and Sweden), the annual prevalence of adult AD based on a patient-reported diagnosis by a physician ranged from 3.3% in Portugal to 9.4% in Germany [4]. In Greece, there are limited available data regarding AD prevalence in the adult population, with the results of a recently conducted nationwide survey indicating that it might range between 2% and 6% depending on the AD definition used [2,5].

There is a growing body of evidence indicating that the impact of AD on health-related quality of life (HRQoL), sleep, daily activities, and work productivity, particularly in patients with more severe disease stages, is substantial [6,7,8,9,10,11]. Children and adults with AD experience a significant burden with a deep impact on HRQoL as well as bullying at school and concerning the domains of daily activities, school, leisure, and personal relationships [12]. Moreover, it seems that AD poses a substantial economic burden since the annual direct and indirect costs reach EUR 6900 and EUR 14,200 per patient, respectively, in Europe [8]. Apart from disease severity, other factors found to be associated with an increased risk of the humanistic and economic burden of AD are obesity or other chronic inflammatory or mental health disorders such as rheumatoid arthritis, anxiety, and depression [13,14].

In Greece, studies assessing the burden of AD are completely lacking. As such, the objective of this study was to describe the burden of AD in terms of QoL, sleep, social life, work productivity, and resource utilization related to AD and assess the impact of disease severity in those domains. Other exploratory objectives of this study were to identify factors that might independently impact QoL, work productivity, sleep, and social activities.

## 2. Materials and Methods

### 2.1. Study Design and Participants

This publication presents the secondary endpoints of a nationwide cross-sectional survey with a structured questionnaire conducted between 17 June 2021, and 12 July 2021, involving a 2-stage procedure. Stage 1 was designed for the evaluation of AD prevalence in Greece, whereas Stage 2 was for the investigation of AD burden. Eligible to participate were all adults (age ≥18 years) living in Greece. Among those selected and accepted to participate in the present survey, data regarding AD burden were collected only for participants identified with AD. All participants were informed a priori for the purposes of the study, and they were asked to provide their consensus for study participation. The study design and the primary objectives of this study are already published elsewhere [5].

The present study was conducted in accordance with the Declaration of Helsinki and the Greek legislation (Law 2328/1995, Presidential Decree 310/1996, Law 3603/2007, Law 2472/1997, Law 3471/2006), stating that there is no need for ethics approval in telephone and internet surveys such the one presented here (Association of Opinion Polls and Survey Organizations, Law 4624, www.sedea.gr, accessed on 29 August 2019). Therefore, the study protocol was not submitted to the ethical committee of any institution for approval.

### 2.2. Data Collection

Computer-assisted telephone interviewing (CATI) and computer-assisted web interviewing (CAWI) methods were used to collect data using a structured questionnaire. The survey was conducted by the MRB Hellas S.A.

All participants were asked about socioeconomic factors (gender, age, residence, weight, height, family status, education level, occupational status), medical history of comorbidities, and family history of asthma, rhinitis, or allergies. Moreover, all participants replied to a set of screening questions about AD, based on the UK Working Party (UKWP)-modified criteria for self-diagnosis of AD (pruritic skin condition and ≥3 of the following: history of skin crease involvement, a personal history of asthma, history of dry skin during past year, symptoms onset under the age of two) as described in the study of Fuxench et al. Additionally, all participants were asked to report if they had an AD diagnosis by a physician (“do you have an AD diagnosis from a medical expert?”) [6,15,16].

Those identified with AD, either via the patient-reported AD diagnosis by a physician (Expert Diagnosis cohort) question or the UKWP criteria (UKWP cohort), answered questions about smoking (currently smoker, former smoker, no smoker), healthcare resource utilization during the last 12 months (i.e., doctor visits, AD treatment, kind of treatment), impact on sleep (“in the last 12 months, how much is your sleep quality generally affected due to AD?” answered with a 5-Likert scale: none at all, a little, moderate, much, very much) and impact on social life (“in the last 12 months, how often has AD prevented you from responding to your social life?” answered with a 5-Likert scale: never, little, somewhat, much, a great deal). The recently translated Greek version of the validated, patient-derived assessment tool Patient-Oriented Eczema Measure (POEM) was used for the assessment of AD severity [17]. Severity was classified as ‘clear or almost clear skin’, ‘mild eczema’, ‘moderate eczema’, ‘severe eczema’, and ‘very severe eczema’ according to the values of POEM score (range: 0 to 28 with a higher score indicating more severe statuses) [18,19,20]. The impact of AD on QoL was examined with the Dermatology Life Quality Index (DLQI); the DLQI score ranges from 0 to 30, and a higher score indicates more impaired QoL. Based on these scores, the effect in QoL is classified as none, small, moderate, very large, and extremely large [21]. The impact on work productivity and regular activities was addressed with the Work Productivity and Activity Impairment (WPAI) questionnaire; WPAI outcomes are expressed as impairment percentages, with higher numbers indicating greater impairment and less productivity, i.e., worse outcomes. Based on the WPAI questionnaire, four scores were calculated: percent work time missed due to AD (absenteeism), percent impairment while working due to AD: (presenteeism), percent overall work impairment due to AD: (absenteeism plus presenteeism), and percent activity impairment due to AD [22]. The English translation of the study questionnaire is presented in Appendix A.

### 2.3. Statistical Analysis

The linearization-based variance estimators were used to take account of survey weights in computing sampling standard errors. Categorical variables were summarized by relative frequencies (%) and continuous parameters with means and 95% confidence intervals (CI) in the overall population of each cohort (UKWP and Expert Diagnosis) and stratified by AD severity (Clear skin to mild eczema vs. Moderate to very severe eczema). The association of AD severity with sociodemographic and clinical factors was investigated at the univariate level with the Pearson χ2 statistic corrected for the survey design with the second-order correction of Rao and Scott (1984) and converted into an F statistic.

For the assessment of the impact of sociodemographic and clinical factors on QoL (moderate to extremely large effect vs. no effect to small effect based on DLQI score), sleep (moderate to very large effect vs. none to small effect) and social activities (somewhat to a great deal vs. never to little), a multiple logistic regression model was fitted, while for the work productivity (overall work impairment), a multiple generalized linear model (GLM) with a log link function under the gaussian distribution was fitted. Gender, age, and body mass index (BMI) were included in all multiple models irrespective of their association with the dependent variables in the univariate models: comorbidities, smoking, disease severity, and time since diagnosis were investigated for inclusion in the multiple models, where applicable: those with *p* ≤ 0.15 in the univariate level were considered for inclusion, after checking for inter-correlation. All models were corrected for the survey design. Logistic regression results were presented with odds ratios (OR) and 95% CIs, while for the GLM, the exponential of beta coefficients (exp(b)) and their 95% CI were presented.

All tests were carried out with a 5% α-error rate. Data cleaning, data manipulation and data analysis were conducted using the statistical software Stata (version 14.2, 2017, STATA Corp).

## 3. Results

### 3.1. Participants’ Information

In total, 1487 responded to the AD screening questions, and 69 and 349 participants of the survey were identified with AD based on the UKWP criteria and patient-reported diagnosis by a physician, respectively. Among those, 53.0% and 25.8%, respectively, were classified with moderate to very severe eczema based on the POEM severity classification tool.

More than half (54.5%) of the participants in the UKWP cohort were 30 to 49 years old, while 61.3% of them were female. Most responders were overweight or obese (58.9%), 52.2% were married or cohabitating, 61.7% had received a bachelor’s or higher degree, and 69.8% were employees or self-employed. These characteristics were well-balanced between participants identified with clear skin to mild eczema and those with moderate to very severe eczema. Similar distributions of the sociodemographic characteristics were observed in the Expert Diagnosis cohort (Table 1).

The most common comorbidities for responders in the UKWP cohort were allergies (57.3%) and asthma (42.3%), while 71% reported a family history of AD, asthma, or rhinitis. Almost half (49.2%) were current smokers, who preferred smoking conventional cigarettes. No discrepancies were observed between those with clear skin to mild eczema and those with moderate to very severe eczema.

In the Expert Diagnosis cohort, the most common comorbidities reported were allergies (defined as allergic rhinitis, conjunctivitis, and food allergy) (38.1%), GI problems, and seasonal rhinitis (both in 17.2%), while 54% reported a family history of AD, asthma, or rhinitis. Participants with asthma (30.3% vs. 10.6%; *p* < 0.001), allergies (49.4% vs. 34.0%; *p* = 0.012), heart failure (5.9% vs. 1.3%; *p* = 0.023), and family history of AD, asthma, or rhinitis (68.5% vs. 48.7%; *p* = 0.002) were more prevalent in those with moderate to very severe eczema compared to those with clear skin to mild eczema. Moreover, those with moderate to very severe eczema were more likely to be smokers compared to those with clear skin or mild eczema (48.6% vs. 30.6%; *p* = 0.001) (Table 2).

### 3.2. Impact on QoL, Sleep and Social Life

The impact of AD on QoL was statistically significantly higher among those with moderate to very severe eczema in both study cohorts. In the UKWP cohort, the effect of AD was moderate to extremely large in 84.3% of participants with moderate/very severe eczema (vs. 55.7% in those with clear skin/mild eczema; *p* = 0.016), while in the Expert Diagnosis cohort, it was 72.2% (vs. 22.8%; *p* < 0.001). Similarly, higher disease severity was associated with a higher impact on sleep and social life in both study cohorts. (Figure 1) Considering the UKWP cohort and based on the DLQI subscales, those with moderate/severe disease reported a higher effect of AD in their skin-related QoL, compared to those with clear skin/mild eczema, during their leisure time, during work or school, and during the application of AD treatment. This effect was higher in those with moderate/severe disease in the Expert Diagnosis cohort, compared to clear skin/mild eczema for all DLQI subscales (Symptoms and Feelings, Daily Activities, Leisure, Work and School, Personal Relationships, Treatment) (Appendix A).

Disease severity remained a significant determinant of QoL, sleep, and social life after adjustment for other demographic, socioeconomic, and medical factors in both study cohorts. More specifically, participants in the UKWP and Expert Diagnosis cohorts with moderate to very severe eczema had 5.4 and 7.9 times, respectively, higher odds to have moderate to extremely large impact on their QoL compared to those with clear skin/mild eczema. (Appendix A). Additionally, in participants of the UKWP and Expert Diagnosis cohorts, the corresponding odds of moderate to high impact on sleep were 5.3 and 5.6 times, respectively, higher, while the odds of somewhat to great deal impact on social life were 6.6 and 7.1 times, respectively, higher among those moderate to very severe eczema compared to those with clear skin/mild eczema (Appendix A).

### 3.3. Impact on Work Productivity and Overall Activities

A total of 68.8% and 62.0% of UKWP and Expert Diagnosis participants were working at the time of data collection. Among them, absenteeism reached 18.3% and 6%, respectively; in the Expert Diagnosis cohort, absenteeism was significantly higher for those with moderate to very severe eczema (11.5% work time lost) compared to those with clear skin/mild eczema (3.8% work time lost; *p* = 0.029). Overall work impairment was high in both clear skin/mild eczema (32.7%) and moderate to very severe eczema (48.5%) participants of the UKWP cohort, while among Expert Diagnosis participants, it was significantly higher among those with moderate to very severe eczema (31.2%) relative to those with clear skin/mild eczema (11.9%; *p* < 0.001). In both cohorts, lost productivity appeared to be driven by presenteeism, which was higher among those with moderate to very severe eczema. Activity impairment was statistically significantly higher in participants with moderate to very severe eczema (39.3%; 30.6%, respectively) relative to those clear skin/mild eczema (21.2% and 10.2%, respectively) for both UKWP and Expert Diagnosis cohorts (Table 3).

As before, disease severity remained a significant determinant of overall work productivity after adjustment for other demographic, socioeconomic, and medical factors in both study cohorts. More specifically, participants in the UKWP and in Expert Diagnosis cohorts with moderate to very severe eczema were expected to have 51% and 120%, respectively, higher work productivity losses compared to those with clear skin/mild eczema. (Appendix A).

### 3.4. Healthcare Resource Utilization

A high percentage of participants of the UKWP (77.8%) and Expert Diagnosis (53.8%) cohorts sought medical help during the last 12 months. In both cohorts, those with moderate to very severe eczema (UKWP: 88.6%; Expert Diagnosis: 80.3%) were more likely to have visited their doctor at least 1 time during the last 12 months compared to those with clear skin/mild eczema (UKWP: 64.4%, *p* = 0.024; Expert Diagnosis: 44.3%, *p* < 0.001). At least four out of five were under treatment for AD in the UKWP cohort, while more than half in the Expert Diagnosis cohort, in which those with moderate to very severe eczema, were more likely to be under treatment. Topical treatment was the most commonly used option, irrespective of cohort and AD severity, followed by oral systemic treatment and nutrition (Table 4).

**Table 3 jcm-11-04777-t003:** Impact of AD in work productivity by study cohort and AD severity the last 7 days.

	UKWP				Expert Diagnosis			
	Overall	Clear or almost Clear Skin/Mild Eczema	Moderate to Very Severe Eczema		Overall	Clear or almost Clear Skin/Mild Eczema	Moderate to Very Severe Eczema	
Effect in Work Productivity Due to AD	N = 69	N = 32	N = 36	*p*	N = 349	N = 252	N = 88	*p*
Work time missed due to AD	N = 43	N = 20	N = 22		N = 172	N = 123	N = 48	
Percent (%), mean (95% CI)	18.3 (10.0, 26.6)	13.70 (2.57, 24.83)	23.46 (11.05, 35.87)	0.244	6.0 (3.4, 8.5)	3.84 (1.45, 6.24)	11.50 (5.07, 17.91)	0.029
Impairment while working	N = 45	N = 20	N = 24		N = 182	N = 127	N = 53	
Percent (%), mean (95% CI)	27.9 (20.5, 35.4)	20.88 (10.75, 31.00)	35.16 (25.11, 45.20)	0.050	14.3 (11.3, 17.3)	9.30 (6.58, 12.02)	26.80 (19.77, 33.83)	<0.001
Overall work impairment	N = 43	N = 20	N = 22		N = 171	N = 122	N = 48	
Percent (%), mean (95% CI)	39.9 (31.0, 48.8)	32.66 (20.14, 45.19)	48.54 (36.87, 60.21)	0.068	17.3 (13.5, 21.0)	11.89 (8.25, 15.53)	31.19 (22.78, 39.60)	<0.001
Activity impairment	N = 68	N = 31	N = 36		N = 341	N = 246	N = 88	
Percent (%), mean (95% CI)	30.4 (23.9, 36.9)	21.15 (13.04, 29.25)	39.34 (30.25, 48.42)	0.004	16.0 (13.6, 18.4)	10.23 (8.00, 12.46)	30.59 (25.03, 36.16)	<0.001

UKWP: UK Working Party; AD: atopic dermatitis. All *p*-values displayed were derived by a two-sample *t*-test. POEM score could not be estimated for 1 and 9 participants in the UKWP and Expert diagnosis cohort, respectively, due to missing values in the participants’ replies.

## 4. Discussion

A small number of cross-sectional studies have been performed worldwide in the last decade investigating the prevalence of AD and the burden that AD incorporates in patients’ life, but none of them contain information for Greece.

In a cross-sectional study performed by Fuxench et al. in the US, it was found that 37% of AD patients with moderate disease and 68% of those with severe AD had a moderate to large impact on their QoL based on the DLQI questionnaire [6]. The high burden of AD in QoL was identified in our results as well, although in our case, the effect was higher as more than 84% of the participants with moderate-to-severe AD had impaired QoL. Simpson et al. (2016) reported baseline data about the QoL of moderate-to-severe AD patients from a phase 2b clinical trial. In their study, the mean (SD) DLQI score was 14.3 (7.0), indicating the very large impact of AD on patients’ lives [11]. In the AD-AWARE study, the mean DLQI scores were higher among patients with moderate/severe vs. those with mild disease (9.2 vs. 2.9) [23]. In our case, the mean score was 12.7 and 10.1 in moderate-to-severe AD responders identified with the UKWP criteria or an expert diagnosis, respectively, indicating an important burden as well. Lindberg et al. based on a nationwide epidemiological survey conducted in Sweden reported higher impairments in the QoL of patients with severe eczema versus those with mild eczema. This difference based on AD severity was identified in our results, although QoL was assessed through a different tool [7]. As no QoL data are available for AD in Greece, we compared our results with those of Rigopoulos et al. (2018), a multicenter, cross-sectional study that included 40 sites across Greece in adult, moderate/severe psoriatic patients. In this study, the mean (SD) DLQI score reported was 6.6 (6.5), which was lower than our findings for moderate/severe AD [24]. This difference should be interpreted cautiously due to the different study methods and populations, although it is in accordance with the findings of Egeberg et al., who found a higher self-reported burden of AD compared to the burden of psoriasis, including HRQoL in terms of DLQI score [25].

Girolomoni et al., based on a cross-sectional study conducted in five European countries (France, German, Italy, Spain, and the UK), reported that 61.6% of the moderate-to-severe AD patients had sleep difficulties, a percentage comparable with the 68.7% of moderate/severe AD responders in our study who reported at least a moderate effect in their sleep due to AD. It should be noted that the AD severity definition in Girolomoni et al. was based on the DLQI questionnaire [8]. Additionally, sleeping problems were associated with AD severity in the AD-AWARE study and in a cross-sectional study recently performed in Sweden. More specifically, those with severe AD had almost 8 times higher probability to experience severe sleeping problems compared to those with mild AD, a result aligned with ours, as those with moderate-to-severe AD had more than 5 times increased odds to experience sleeping problems compared to those with mild AD [23,26].

Based on our results, for those in the UKWP cohort who were working, the mean (95% CI) overall work impairment due to AD was 40% (31%, 49%). In the 2013 National Health and Wellness Survey (NHWS) conducted in the US, the overall work impairment reported was 30%, indicating a comparative burden in our results [10]. Disease severity in that study was an effect modifier for absenteeism, overall work impairment, and activity impairment, with those in moderate/severe stages experiencing higher productivity losses compared to mild AD. In our case, for those in the UKWP cohort only, activity impairment reached statistically significant differences between AD severity stages, while among those in the Expert Diagnosis cohort, all productivity loss components were higher for those in moderate/severe disease. Girolomoni et al. reported rates of overall work impairment that ranged from 32% in the UK to 59% in Spain among those with a moderate and very large effect in their QoL based on DLQI, and from 68% in the UK to 74% in Italy among those with an extremely large effect in their QoL based on DLQI. Disease severity was incorporated with a different tool in our study, and the rate of overall work impairment reached 49% and 31% in those with moderate/severe AD in the UKWP and Expert Diagnosis cohorts, respectively [8].

Whiteley et al. reported a mean of 5.6 and 3.4 outpatient visits in the prior 6 months for subjects with and without AD in the US, respectively, irrespective of the reason for the visit; the crude difference (~2 visits) of these two estimations could be attributed to the additional visits an AD patient would seek due to their disease; the mean number of visits due to AD during the last 12 months in our study ranged from 1 to 3 and is slightly lower but comparable with the results of Whiteley et al. [10]. In the same study, the number of visits due to any reason did not differ between AD severity groups, although in our case, for the Expert Diagnosis cohort, which had an equivalent AD definition, the number of visits due to AD was higher for those with moderate/severe eczema. The number of healthcare provider visits in the past 6 months was reported in the study of Girolomoni et al. as well; the mean ranged from ~6 visits in the UK to ~12 in Germany but could not be compared with our results, as this number refers to visits due to any reason.

This study has several strengths. The data collection was performed via telephone interviews and online questionnaires, reaching several age groups with specific preferences (i.e., younger ages prefer internet-based surveys, while more aged subjects are more approachable via telephone), resulting in a representative of the Greek population study for the AD prevalence estimation. The definition of self-reported AD was based on several criteria, the UKWP criteria and a patient-reported expert diagnosis already used in other publications, allowing for a more accurate comparison of results across studies. For the evaluation of important outcomes such as AD severity, impact on QoL, and work productivity, well-known and validated PROs were used. The use of PROs in adult AD is acknowledged by several researchers, as they can support the procedure of decision making [27,28,29]. However, our study has its limitations. Even though the availability of PRO tools is increasing, only POEM and DLQI were included in our study for brevity, due to the telephone and web nature of the survey. Our explorative, cross-sectional study was limited by the absence of a control group, while the relatively small sample size for the UKWP cohort limits the generalizability of our findings about the AD burden. The impact of AD on sleep and social life was collected with study-specific questions, consisting of one item each and constructed for the purposes of this study, setting limits to the reliability of these outcomes. Selection bias may have occurred from the potential differences between subjects who agreed to participate and those who did not. Another limitation is recall bias or reporting bias (participants may have answered questions based on what they perceive to be expected from them). Moreover, the use of modified UKWP criteria due to the lack of collection of the visible dermatitis criterion could introduce bias.

## 5. Conclusions

A significant burden was observed for patients with AD in Greece, which was associated with the disease severity, as those in moderate-to-severe AD are experiencing higher impairments in their QoL, work productivity, sleep, and social activities. Acknowledging this burden is the first step toward applying healthcare decisions, which will benefit AD patients in Greece and reduce the burden on society.

## Figures and Tables

**Figure 1 jcm-11-04777-f001:**
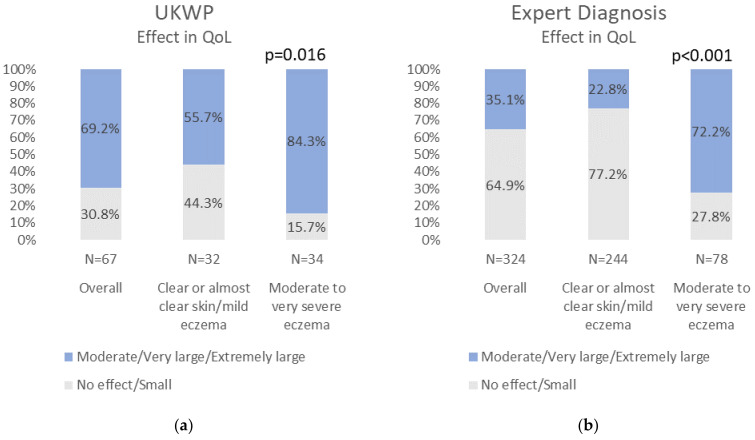
Impact of AD by study cohort and AD severity (**a**) in QoL the last week in UKWP; (**b**) in QoL the last week in Expert Diagnosis cohort; (**c**) in sleep in UKWP the last 12 months; (**d**) in sleep in Expert Diagnosis cohort the last 12 months; (**e**) in social life in UKWP the last 12 months and (**f**) in social life in Expert Diagnosis cohort the last 12 months. UKWP: UK Working Party; AD: atopic dermatitis; QoL: quality of life. All *p*-values displayed were derived by Pearson χ2 statistic corrected for the survey design with the second-order correction of Rao and Scott (1984) and converted into an F statistic.

**Table 1 jcm-11-04777-t001:** Demographic profile of study participants by study cohort and AD severity.

	UKWP				Expert Diagnosis			
	Overall	Clear Skin to Mild Eczema	Moderate to Very Severe Eczema		Overall	Clear Skin to Mild Eczema	Moderate to Very Severe Eczema	
Demographics–General Profile	N = 69	N = 32	N = 36	*p*	N = 349	N = 252	N = 88	*p*
Gender, %			*p* = 0.774					
Female	61.27	62.44	58.98	0.774	63.19	63.42	62.28	0.849
Male	38.73	37.56	41.02		36.81	36.58	37.72	
Age group, %								
18–29	25.80	24.31	24.71		21.71	20.99	21.57	
30–39	33.44	24.89	42.11		28.34	24.99	38.62	
40–49	21.04	30.27	13.53	0.470	18.33	17.83	19.34	0.076
50–59	8.64	6.25	11.05		16.36	19.12	8.88	
60–69	5.89	6.68	5.38		8.28	8.81	7.60	
70+	5.19	7.60	3.23		6.98	8.27	3.98	
Residence, %								
Athens	42.91	31.09	54.79		31.60	30.34	35.15	
Thessaloniki	5.85	9.51	2.78		11.68	12.65	7.82	
Urban area	37.97	37.20	36.64	0.127	42.28	43.19	40.51	0.566
Semi-urban area	2.82	6.10	0		7.78	6.77	10.28	
Rural area	10.45	16.08	5.79		6.67	7.05	6.23	
BMI, %								
Underweight	1.58	0	3.04		3.14	3.14	2.38	
Normal/healthy weight	39.48	37.01	39.72	0.685	44.74	45.24	45.52	0.609
Overweight	42.01	41.88	43.49		35.14	36.52	31.39	
Obese	16.92	21.11	13.76		16.98	15.10	20.72	
Marital status, %					N = 343	Ν = 248	Ν = 87	
Unmarried	40.37	33.69	47.60		37.62	36.13	38.48	
Married/cohabitation	52.20	60.09	43.64		56.97	58.47	55.59	
Divorced	7.44	6.21	8.77	0.409	3.72	3.56	4.53	0.939
Widow/er	0	0	0		1.69	1.85	1.40	
Occupational status, %	N = 67	N = 31	N = 35		N = 344	N = 249	N = 87	
Freelancer/self-employed	21.12	16.83	25.62		17.53	16.25	20.57	
Employee	48.66	50.51	45.32		43.65	42.64	48.28	
Unemployed	5.77	9.37	2.77		10.44	10.79	6.90	
Retired	6.78	7.83	6.08	0.899	9.13	10.01	7.43	0.452
Student	10.27	9.53	11.27		8.31	9.52	4.44	
Household	4.34	3.11	5.58		8.89	9.19	8.84	
Other	3.05	2.82	3.36		2.05	1.59	3.54	
**Education, %**	N = 68	N = 31	N = 36		N = 346	N = 249		
Primary school	3.35	7.38	0		2.57	2.70	2.46	
Lower 2^ndary^ education	0	0	0		2.63	2.09	3.27	
Upper 2^ndary^ education	34.96	32.07	38.57	0.415	29.95	27.30	35.99	0.436
Bachelor’s degree	44.46	44.80	42.38		46.32	47.34	44.76	
Master’s/Doctoral degree	17.23	15.76	19.05		18.54	20.59	13.52	

UKWP: UK Working Party; BMI: body mass index. All *p*-values displayed were derived by Pearson χ2 statistic corrected for the survey design with the second-order correction of Rao and Scott (1984) and converted into an F statistic. POEM score could not be estimated for 1 and 9 participants in the UKWP and Expert diagnosis cohort, respectively, due to missing values in the participants’ replies.

**Table 2 jcm-11-04777-t002:** Medical profile and smoking habits of study participants by study cohort and AD severity.

	UKWP				Expert Diagnosis			
	Overall	Clear or almost Clear Skin/Mild Eczema	Moderate to Very Severe Eczema		Overall	Clear or almost Clear Skin/Mild Eczema	Moderate to Very Severe Eczema	
Clinical Profile & Habits	N = 69	N = 32	N = 36	*p*	N = 349	N = 252	N = 88	*p*
Time since AD diagnosis, years					N = 328	N = 236	N = 85	
Mean (95% CI)	NA	NA	NA		17.6 (16.1, 19.1)	18.3 (16.5, 20.1)	16.1 (13.3, 18.8)	0.184
Comorbidities, %						N = 249	N = 87	
None	22.67	24.42	23.06		33.67	37.03	24.04	
One	23.49	25.26	22.69	0.981	32.58	32.62	33.54	0.098
Two	28.39	27.31	30.28		19.68	17.25	26.39	
Three or more	25.45	23.01	25.19		14.07	13.11	16.04	
Comorbidities, %					N = 345	N = 249	N = 87	
Asthma	42.28	46.88	36.32	0.386	15.24	10.55	30.27	<0.001
COPD	1.27	0	2.44	0.349	2.29	1.21	4.51	0.065
Allergies (Allergic rhinitis, conjunctivitis, food allergies)	57.34	52.43	60.31	0.520	38.14	34.01	49.36	0.012
GI problems	21.12	16.92	25.54	0.407	17.20	15.88	20.54	0.323
Seasonal rhinitis	21.70	18.97	24.83	0.567	17.18	17.20	15.48	0.711
Diabetes	10.49	7.57	13.42	0.467	6.45	5.35	10.25	0.112
Hypertension	12.36	7.30	14.01	0.405	14.96	17.03	10.51	0.152
Heart failure	4.47	0	8.57	0.964	2.42	1.30	5.88	0.023
					N = 346	N = 251	N = 86	
Family history of AD, rhinitis, asthma, %	71.04	59.65	80.22	0.071	54.04	48.67	68.51	0.002
Smoking habits, %					N = 347	N = 251	N = 87	
Smoker	49.21	37.32	58.13		35.43	30.59	48.56	
Former smoker	28.82	37.36	22.17	0.225	28.40	27.39	30.79	0.001
No smoker	21.97	25.32	19.71		36.16	42.02	20.65	
Smoking products, % ^1^	N = 34	N = 12	N = 21		N = 124	N = 78	N = 42	
Cigarettes	85.02	83.89	90.40	0.587	87.28	90.07	81.03	0.163
Electronic cigarettes	37.39	41.94	36.88	0.779	24.96	18.69	38.70	0.018
Heated tobacco products	38.30	24.06	42.97	0.288	20.99	18.39	27.68	0.238

UKWP: UK Working Party; AD: atopic dermatitis; COPD: chronic obstructive pulmonary disease; GI: gastrointestinal; NA: not applicable. All *p*-values displayed were derived by Pearson χ2 statistic corrected for the survey design with the second-order correction of Rao and Scott (1984) and converted into an F statistic. POEM score could not be estimated for 1 and 9 participants in the UKWP and Expert diagnosis cohort, respectively, due to missing values in the participants’ replies. ^1^ Participants could choose more than one option.

**Table 4 jcm-11-04777-t004:** Healthcare resource utilization due to AD by study cohort and AD severity the last 12 months.

	UKWP				Expert Diagnosis			
	Overall	Clear or almost Clear Skin/Mild Eczema	Moderate to Very Severe Eczema		Overall	Clear or Almost Clear Skin/Mild Eczema	Moderate to Very Severe Eczema	
	N = 69	N = 32	N = 36	*p*	N = 349	N = 252	N = 88	*p*
	N = 68	N = 20	N = 36		N = 329	N = 240	N = 82	
At least 1 doctor visit due to AD, %	77.80	64.44	88.61	0.024	53.83	44.31	80.30	<0.001
Number of visits due to AD, mean (95% CI)	2.88 (1.72, 4.04)	2.88 (0.66, 5.11)	2.94 (1.87, 4.01)	0.695 ^‡^	1.41 (1.12, 1.70)	0.99 (0.67, 1.31)	2.64 (2.04, 3.24)	<0.001 ^‡^
Received treatment for AD					N = 342	N = 248	N = 86	
Yes, %	81.43	75.80	85.82	0.301	58.86	54.22	76.55	<0.001
Kind of treatment, %	N = 56	N = 24	N = 31		N = 201	N = 134	N = 66	
Nutrition	21.24	4.19	31.62	0.015	15.73	10.31	27.03	0.003
Topical treatment	80.79	83.66	77.82	0.592	87.14	89.61	81.91	0.130
Phototherapy	5.20	0	9.46	0.118	2.95	0	9.03	<0.001
Systemic treatment (oral)	31.71	29.22	34.86	0.662	22.18	17.78	31.51	0.030
Antibiotics	10.60	12.99	9.13	0.651	4.51	3.06	7.55	0.159
Immunotherapy	5.01	0	9.13	0.120	2.35	0	7.19	0.001
Biologic agents	8.78	4.14	12.76	0.276	2.95	1.42	6.11	0.065

UKWP: UK Working Party; AD: atopic dermatitis. All *p*-values displayed were derived by Pearson χ2 statistic corrected for the survey design with the second-order correction of Rao and Scott (1984) and converted into an F statistic, except otherwise specified. POEM score could not be estimated for 1 and 9 participants in the UKWP and Expert diagnosis cohort, respectively, due to missing values in the participants’ replies. ^‡^
*p*-value displayed by a two-sample *t*-test.

## Data Availability

Data will be available by the authors upon reasonable request.

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
