# Peer review of "Burden of Atopic Dermatitis in Adults in Greece: Results from a Nationwide Survey"

_jcm, 2022, doi:10.3390/jcm11164777_

Round 1

Reviewer 1 Report

The article “Burden of atopic dermatitis in adults in Greece: Results from a 2 nationwide survey” describes the results of a survey conducted in Greece to identify the burden of AD in terms of work productivity, QoL, sleep, and social activities.

 I have some suggestions for the authors:

Please, in the introduction, underline that not only adults but also children with AD experience a significant burden with a deep impact on HRQoL as well as bullying at school and concerning the domains of daily activities, school, leisure, and personal relationships and add the following reference: Atopic Dermatitis and Patient Perspectives: Insights of Bullying at School and Career Discrimination at Work. J Asthma Allergy 2021; 21; 14: 919-928.

 In the article, you consider POEM and DLQI, as tools to define the severity of AD and the quality of life perceived by the patients. POEM and DLQI belong to patient-reported outcomes (PROs), self-assessing validated questionnaires, reported directly by people about their status of health condition, symptoms, and quality of life through. However, PROs include also Night Time Itch Scale - NTIS, Pruritus and Symptoms Assessment for Atopic Dermatitis - PSAAD, Patient Global Assessment - PGA, SCORAD, Hospital Anxiety, and Depression Scale - HADS, and EuroQoL 5-Dimension 5-Level - EQ-5D-5L, as reported in these recent articles:

-          Patient-reported outcomes in adult atopic dermatitis are useful in both clinical trials and real-life clinical practice. J Eur Acad Dermatol Venereol. 2022 Mar;36(3):326-327. doi: 10.1111/jdv.17956. PMID: 35174911.)

-          Thyssen JP, Yosipovitch G, Paul C et al. Patient-reported outcomes from the JADE COMPARE randomized phase 3 study of abrocitinib in adults with moderate-to-severe atopic dermatitis. J Eur Acad Dermatol Venereol 2021 Nov 15. doi: 10.1111/jdv.17813. Epub ahead of print

Perhaps you only consider POEM and DLQI for brevity, due to the telephone and web nature of the survey. Please, in the discussion, clarify the role of the PROs and emphasize that the use of POEM and DLQI only is a limitation of your study and add the reference reported.

Reviewer 2 Report

Abstract:

Please clarify what you mean with UKWP cohort and Expert Diagnosis cohort, that does not become very clear from reading the abstract.

Introduction

Line 32,36: The annual prevalence of doctor-diagnosed AD rages from …

Line 40-41: self-reported diagnosis by a physician?? Maybe patient-reported diagnosis b physician

Line 51-52: associated with an increased risk of

Materials and Methods

You might provide the questionnaire as a supplement

2.1 Study design and participants

Line 61: it is not completely clear what the first sentence intends to mean. I think what you mean is ‘secondary endpoints’. Please declare, that you already published parts of the very same survey elsewhere.

You mention informed consent at the end of your manuscript, but not in section 2. Did the patients sign an informed consent form?

2.2 Data collection:

How were the individuals to take part in the study chosen? How did you get their telephone numbers?

Line 132: what does without correction for multiplicity mean? w/o correction for multiple hypotheses? If yes, why did you not correct for it?

Results

How many did participate in the study (not only AD patients)?

How do you explain the big gap between Expert diagnosis and UKWP-based classification?

Lines 159-160: please specify how you define different groups (e.g. rhinitis seems to occur in allergies and as a separate co-morbidity – why? Do you account for that?)

Figure 1: bad quality, non-consistent capital/non-capital writing

Reviewer 3 Report

The article as such is interesting, although it may be somehow difficult to comprehend in some places because of long and convoluted sentences.

The references are not included in the text according to editorial policy.

Author Response

Thank you for your comments, we changed the formating of references to the MDPI format.